# Children with Heiner Syndrome: A Single-Center Experience

**DOI:** 10.3390/children8121110

**Published:** 2021-12-01

**Authors:** Ji Young Lee, Mireu Park, Jae Hwa Jung, Soo Yeon Kim, Yoon Hee Kim, Seung Min Hahn, Seung Kim, Mi-Jung Lee, Hyo Sup Shim, Myung Hyun Sohn, Kyung Won Kim, Min Jung Kim

**Affiliations:** 1Department of Pediatrics, Severance Hospital, Institute of Allergy, Yonsei University College of Medicine, Seoul 03722, Korea; sophielee0319@yuhs.ac (J.Y.L.); QKRALFM27@yuhs.ac (M.P.); jung627B@yuhs.ac (J.H.J.); Sophi1@yuhs.ac (S.Y.K.); yhkim@yuhs.ac (Y.H.K.); mhsohn@yuhs.ac (M.H.S.); kwkim@yuhs.ac (K.W.K.); 2Department of Pediatric Hematology-Oncology, Yonsei Cancer Center, Yonsei University College of Medicine, Seoul 03722, Korea; bluenile88@yuhs.ac; 3Division of Gastroenterology, Hepatology and Nutrition, Department of Pediatrics, Severance Hospital, Yonsei University College of Medicine, Seoul 03722, Korea; kim.seung@cchmc.org; 4Department of Radiology, Research Institute of Radiological Science and Center for Clinical Imaging Data Science, Yonsei University College of Medicine, Seoul 03722, Korea; mjl1213@yuhs.ac; 5Department of Pathology, Yonsei University College of Medicine, Seoul 03722, Korea; shimhs@yuhs.ac; 6Department of Pediatrics, Yongin Severance Hospital, Yonsei University College of Medicine, Yongin-si 16995, Korea

**Keywords:** cow’s milk allergy, pulmonary hemosiderosis, pediatric

## Abstract

Heiner syndrome is a rare cause of pulmonary hemosiderosis in children that is triggered by cow’s milk allergy. Herein, we describe our experience with three recent cases of Heiner syndrome with diverse clinical courses. We recommend that clinicians should consider the possibility of Heiner syndrome in children who exhibit characteristics of idiopathic pulmonary hemosiderosis.

## 1. Introduction

Idiopathic pulmonary hemosiderosis (IPH) is a nonimmune-mediated type of diffuse alveolar hemorrhage syndrome, which is commonly diagnosed in infancy and early childhood. It is typically characterized by a triad of iron deficiency anemia, hemoptysis, and diffuse pulmonary infiltrates; however, in children, symptoms and signs are nonspecific and this leads to misdiagnosis [1]. The diagnosis of IPH is challenging and requires a multifaceted evaluation after excluding secondary causes such as infection and autoimmunity. Pulmonary hemosiderosis is pathologically confirmed by the presence of hemosiderin-laden macrophages in alveoli by broncho-alveolar lavage or lung biopsy. However, when a patient is diagnosed with IPH, there are additional considerations. Heiner syndrome is a pulmonary hemosiderosis caused by cow’s milk hypersensitivity, which was first described by Heiner in 1962 [2]. It is diagnosed using a combination of clinical, laboratory (including positive precipitating IgG antibodies to cow’s milk), and radiographic evidence. Most importantly, the clinical exacerbation of pulmonary symptoms or chest radiograph findings after cow’s milk consumption is pathognomonic; however, since the cow’s milk provocation test is realistically dangerous and the treatment involves systemic corticosteroids, the effect of cow’s milk restriction alone remains unknown [3].

Herein, we describe our experience with three recent cases of Heiner syndrome presenting with different symptoms and clinical courses.

## 2. Case Reports

We describe three patients who presented with different initial symptoms. The clinical presentations are variable in Heiner syndrome; the patient in Case 1 showed anemia and mild fever, while the patient in Case 2 exhibited hematemesis. The patient in Case 3 was the only one who had the typical symptom of hemoptysis and failure to thrive. As all cases had diffuse pulmonary infiltrates on chest radiographs without evidence of infection, we suspected IPH. They underwent the common differential diagnostic process of ruling out infectious and autoimmune causes shown in Table 1 and did not have meaningfully positive findings. The cases were then pathologically confirmed as pulmonary hemosiderosis by lung biopsy. The first two cases had detectable precipitating IgG antibodies to cow’s milk while the third case was not tested. While this antibody titer may provide helpful clues in diagnosing Heiner syndrome, its role is unclear and it may be present in 1% of asymptomatic healthy children [4]. Although the first patient probably had Heiner syndrome due to an unclear correlation between her recurrence of symptoms and cow’s milk exposure, she was instructed to avoid it and since then, pulmonary hemorrhage has not relapsed. In contrast, pulmonary hemorrhage episodes in the second patient recurred multiple times due to accidental milk consumption. The last patient was confirmed to have Heiner syndrome from pulmonary hemorrhage by the milk provocation test. A comparison of clinical manifestations and diagnostic results is shown in Table 1.

### 2.1. Case 1

Case 1 was a 2-year-old boy who was admitted to the department of hemato-oncology due to pallor without respiratory symptoms or signs including no hemoptysis. Laboratory results revealed severe anemia, and his chest radiograph and chest computed tomography scans revealed pulmonary hemorrhage as the focus of bleeding (Figure 1). The patient was diagnosed with IPH and treated with corticosteroids. His clinical course was uneventful and the corticosteroid dose was gradually tapered after the first month of treatment. However, he was re-admitted due to hemoptysis. Although he had no history of allergy and low levels of specific immunoglobulin (Ig)E to cow’s milk, Heiner syndrome was nevertheless suspected, and milk avoidance was recommended. The patient has been adhering to a strict milk restriction diet and has not had any further hemorrhagic events, and is not taking corticosteroids.

### 2.2. Case 2

Case 2 was a 1-year-old girl who presented with recurrent hematemesis. She was diagnosed with IPH, and systemic corticosteroids and avoidance of cow’s milk were recommended based on our clinical experience with the first case. However, due to multiple episodes of accidental milk ingestion, she experienced repetitive pulmonary hemorrhage despite corticosteroid therapy. Given the exacerbation of clinical symptoms after milk exposure, she was diagnosed with Heiner syndrome. This case demonstrated the importance of corticosteroid therapy and strict milk restriction. At 2 years after diagnosis, the patient underwent an oral milk provocation test for 5 days, and she showed no symptoms or signs of hemorrhage. 

### 2.3. Case 3

Lastly, case 3 was a 2-year-old boy who presented with hemoptysis. Clinical investigations were performed to rule out pulmonary tuberculosis and other infectious causes, and these all came back negative. Two years later, he was hospitalized twice for pneumonia while living abroad and probably was accompanied by pulmonary hemorrhage due to first onset of anemia. Hemoptysis recurred at the age of 3; therefore, he underwent a comprehensive work-up including lung biopsy, which confirmed pulmonary hemosiderosis. Although the patient had no history of cow’s milk allergy, milk avoidance and systemic corticosteroids were initiated. Oral milk provocation was attempted 1 year later by introducing cow’s milk and dairy products such as cheese and ice cream every day for 1 week; this led to increased sputum and pulmonary infiltrates on the chest radiograph (Figure 2). As a result, this patient was diagnosed with Heiner syndrome.

## 3. Discussion

IPH is a rare clinical condition in children with variable clinical manifestations; therefore, clinical suspicion is essential to ensure appropriate diagnostic work-up. Early diagnosis is critical to initiate proper treatment and to prevent recurrent hemorrhagic episodes that lead to pulmonary fibrotic changes, and thus to increase the survival rate of affected patients. Of note, when IPH is diagnosed by exclusion, Heiner syndrome should be suspected since avoidance of cow’s milk may improve the symptoms. 

The pathophysiology of IPH has not been delineated, but the combination of genetic predisposition with environmental triggers such as toxin-producing fungus (*Stachybotryschatarum*) has been proposed [5,6]. Moreover, there are many hypotheses regarding cow’s milk hypersensitivity in Heiner syndrome such as immune complex formation against cow’s milk protein, deposition of complement, and involvement of immunoglobulin, fibrin, and milk protein, and so on, but the exact mechanism remains unclear. The underlying mechanism of the association between cow’s milk consumption and pulmonary hemorrhage in patients with Heiner syndrome is unclear, but the possibility of milk aspiration in children has been suggested [3]. Although diagnosis is challenging, there are some clues. First, a history of cow’s milk allergy and testing for specific IgE and IgG to cow’s milk could be helpful [4]. Second, the clinical course of dramatic clinical and radiological improvement with systemic corticosteroids and with cow’s milk restriction within 1–2 weeks could also support the diagnosis of Heiner syndrome. In addition, an oral milk provocation test can be considered, as in our third patient. For instance, in a study by Moissidis et al., cow’s milk was restricted in eight patients for 2 years, and there was no recurrence within this period. However, two patients developed symptoms when cow’s milk was reintroduced [3].

Based on a non-systematic review, the first structured diagnostic criteria for Heiner syndrome were proposed in 2021. If a patient has pulmonary symptoms and pulmonary infiltrate on chest x-ray or pathology-confirmed pulmonary hemosiderosis, which resolves after milk removal, then he or she “probably” has Heiner syndrome. Additionally, if the symptoms recur after milk reintroduction, then there is higher likelihood of Heiner syndrome; thus, the patient’s findings that are “convincing” suggest Heiner syndrome, but the provocation or reintroduction is not mandatory for the diagnosis [7]. The authors here emphasize the importance of the differential diagnosis of Heiner syndrome as the prognosis and management are different from those of IPH. First of all, as Heiner syndrome is reversible with cow’s milk restriction, the prognosis is favorable. 

On the other hand, previous studies reported that the long-term outcome of IPH was poor with a mortality rate as high as 50%. The main cause of death was chronic respiratory failure related to pulmonary fibrosis or pulmonary hemorrhage [8]. The mean survival time after diagnosis of IPH was 2.5 years in older reports [9], but recent studies have reported 5-year survival rates above 80% [10]. A 10-year follow-up study of 15 IPH cases in 2000 revealed that most patients (12/15) treated with systemic corticosteroids had mild or no respiratory symptoms and five of them were able to taper off treatment within 10 years. Three cases had severe symptoms including chronic respiratory failure and were awaiting lung transplantation at the time of publication, and the mean follow-up period was 17.2 years [10]. Due to this opposing prognosis between IPH and Heiner syndrome, it is important to differentiate the two diagnoses.

## 4. Conclusions

We presented rare cases of Heiner syndrome treated at our hospital with different clinical courses. The process of suspecting and diagnosing IPH, and subsequently Heiner syndrome, was time-consuming and challenging. However, it is important to diagnose Heiner syndrome because unlike IPH, which is idiopathic, Heiner syndrome has a reversible cause and patient prognosis may be improved by avoiding cow’s milk and systemic corticosteroids. Thus, we suggest that until fully excluded, Heiner syndrome should be considered in the differential diagnosis of patients with IPH, and early cow’s milk restriction should be initiated. This is especially important since medical treatment alone was insufficient in treating pulmonary hemorrhage, which highlights the need to avoid cow’s milk simultaneously.

## Figures and Tables

**Figure 1 children-08-01110-f001:**
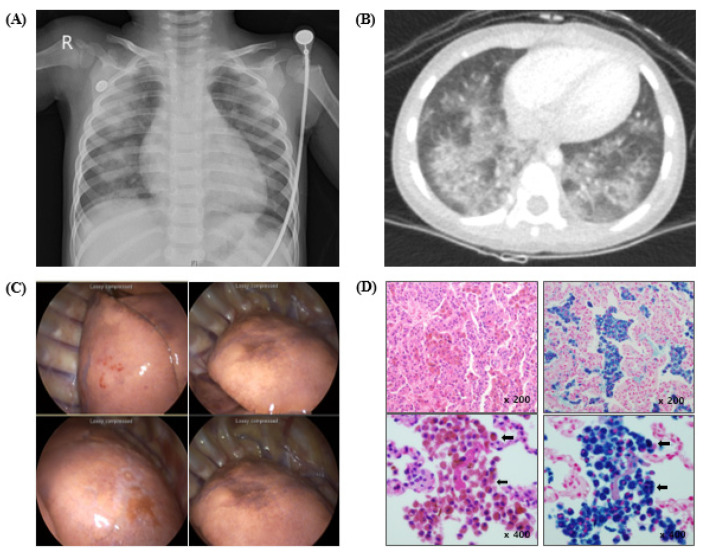
Radiographic and pathologic findings of Case 1 with IPH. (**A**) Initial chest radiograph showing bilateral pulmonary infiltrates. (**B**) Chest computed tomography (CT) revealing diffuse bilateral symmetric alveolar infiltrates. (**C**) Lungs showing brownish discoloration. (**D**) H & E stain (left) and Prussian blue stain (right) (×200, ×400) showing hemosiderin-laden macrophages in alveolar spaces (black arrows).

**Figure 2 children-08-01110-f002:**
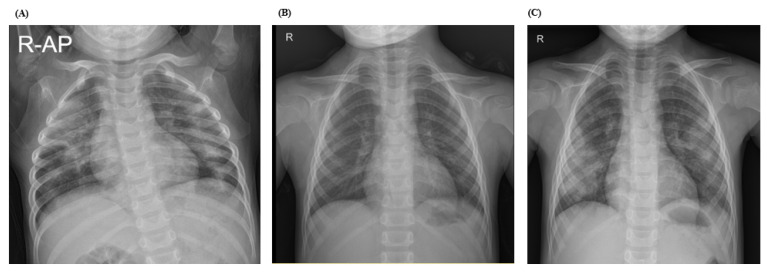
Progress of chest radiograph of Case 3. (**A**) Initial chest radiograph (R-AP) during the first hemoptysis event. (**B**) Resolution of diffuse pulmonary infiltrates after 2 months of systemic corticosteroid treatment and cow’s milk restriction. (**C**) Recurrence of pulmonary infiltrate after cow’s milk provocation test.

**Table 1 children-08-01110-t001:** Initial clinical characteristics of children with Heiner syndrome.

	Case 1	Case 2	Case 3
Sex/Age (years)	M/2.3	F/1.0	M/2.4
Clinical presentation	Pallor	Hematemesis	Hemoptysis
Duration of symptoms	3 days	3 months (3 times)	Recurrent (2018, 2020)
Past History	None	Allergy to milk and egg, atopic dermatitis	Croup, pneumonia
**Imaging Studies**			
Chest radiograph	Multifocal patchy consolidation		
Chest CT	Bilateral, multifocal, and patchy ground glass opacities and consolidations		
**Anemia/Hematologic**			
Hemoglobin (g/dL)	3.1	7.0	11.4
PT/INR (sec)	11.9/1.04	13.1/1.14	12.0/1.05
Serum iron (ug/dL)/TIBC	6/410	30/363	46/355
Transferrin saturation (%)	1	8	15
Ferritin (ng/mL)	29.3	65.6	90
**Microbiology**			
Blood/urine/sputum culture	No growth	No growth	No growth
Respiratory virus/PCP PCR	Negative	Rhinovirus, Coronavirus 229E	Negative
Mycoplasma Ab	1:320	Negative	1:160
EBV/CMV PCR	Negative	Negative	Negative
Aspergillus/Candida Ag	Negative	Negative	Negative
**Immunology**			
ANA/anti-DNA titration	Negative	Negative	1:320/Negative
MPO/PR3(P-ANCA/C-ANCA)	Negative	Negative	Negative
Anti-GBM Ab	Negative	Negative	Negative
C3/C4 (mg/dL)	155/41	95/7	100/11
**Allergy** Total IgE (kU/L)Milk-specific IgE (kU/L)	48.5 0.12	95.5 6.0	8.3 0.03
Milk-specific IgG4 (kU/L)	2.03	1.05	Not tested
**NGS**	Non-pathognomic		
**Echocardiography**	Nonspecific findings		
**Progress**			
Milk avoidance /provocation test	After first recurrence—until now/not done	At the time of diagnosis—until now/done	At the time of diagnosis—until now/done
Duration of follow-up	2 years and 10 months	2 years and 1 month	2 years and 10 months

CT: computed tomography. TIBC: total iron binding capacity. PCP: Pneumocystis pneumonia. PCR: polymerase chain reaction. Ag: antigen. EBV: Epstein-Barr virus. CMV: cytomegalovirus. ANA: Antinuclear antibody. ANCA: antineutrophil cytoplasmic antibody. GBM: glomerular basement. MPO: myeloperoxidase. IgE: immunoglobulin E. NGS: next generation sequencing. PR3: proteinase 3.

## Data Availability

The data presented in this study are available within the article.

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
