# Peer review of "Children with Heiner Syndrome: A Single-Center Experience"

_children, 2021, doi:10.3390/children8121110_

Round 1
Reviewer 1 Report
This is a case report of three cases of Heiner syndrome. Case 2 does not show evidence of pulmonary hemosiderosis. There is no description of precipitating antibodies to cow’s milk, which is essential for diagnosing Heiner syndrome in all cases, and the diagnosis of Heiner syndrome is questionable.
Case 3 does not include details of the milk oral challenge test, and there is no definitive evidence of pulmonary hemorrhage.
The table may not have the correct row layout.
In Figure 2, the resolution is low, and the infiltration shadow is not clearly recognized.
Author Response
[Response to Reviewer 1]
#Point 1.This is a case report of three cases of Heiner syndrome. Case 2 does not show evidence of pulmonary hemosiderosis. There is no description of precipitating antibodies to cow’s milk, which is essential for diagnosing Heiner syndrome in all cases, and the diagnosis of Heiner syndrome is questionable.
Response: Case 2 showed an unusual initial presentation of hematemesis but later symptoms were all hemoptysis. Also, she showed evidence of pulmonary hemosiderosis in lung biopsy pathology. As you have rightfully mentioned, the precipitating antibodies to cow’s milk is helpful in diagnosing Heiner syndrome, I have added the titer to Table 1. Case 1 and 2 were both positive and the third case has not yet been done.
However, while laboratory test may provide a helpful clue in diagnosing Heiner syndrome, it is not pathognomic, and we wanted to emphasize the importance of clinical diagnosis of Heiner syndrome. As also mentioned in a recent journal by Arasi S et al. in 2021 entitled “Heiner Syndrome and Milk Hypersensitivity: An Updated Overview on the Current Evidence,” the proposed diagnostic approach is made probable when (A) pulmonary symptoms and chest x-ray infiltrates or pulmonary hemosiderosis; and (B) resolution after milk restriction; and convincing if also (C) recurrence after milk reintroduction, which our case report also supports. Milk precipitin as well as serum IgE and/or allergic sensitization to milk as positive skin prick test are all tests that may be positive that can hint and favor the diagnosis of Heiner syndrome but if clinically does not exacerbate at the exposure of milk, patient is less likely to be Heiner syndrome. So the tests are some supplementary information to be considered in our point of view, so was not done in Case 3.
#Point 2. Case 3 does not include details of the milk oral challenge test, and there is no definitive evidence of pulmonary hemorrhage.
Response: Milk oral challenge test was not done in a systematic matter, but in fact the patient was given 40cc of cow’s milk and then 100cc the next day, and also dairy products such as cheese and ice cream every day for one week that patient did not exhibit hemoptysis but increased sputum and pulmonary infiltrate on chest radiograph upon outpatient clinic visit. I will add this detail in the main text.(page 4, line 90-92)
“Oral milk provocation was attempted one year later by introducing cow’s milk and dairy product every day such as cheese, ice cream for one week and this led to increased sputum and pulmonary infiltrates on chest radiograph”
#Point 3. The table may not have the correct row layout.
Response: I have made the changes necessary regarding the row layout.
#Point 4. In Figure 2, the resolution is low, and the infiltration shadow is not clearly recognized
Response: The infiltrative shadow of the right middle lung field in the Figure 2c can be observed.
Reviewer 2 Report
Heiner's syndrome is a rare, not-well described disorder in young children. Presentation of 3 cases of the rare disease is very informative.
Most important issue is the correct diagnosis and to improve quality of the report, laboratory diagnostic tests should be discussed such as precipitation IgG antibody.
Author Response
[Response to Reviewer 2]
Heiner's syndrome is a rare, not-well described disorder in young children. Presentation of 3 cases of the rare disease is very informative.
Point #1. Most important issue is the correct diagnosis and to improve quality of the report, laboratory diagnostic tests should be discussed such as precipitation IgG antibody.
Response: Thank you for your helpful comment. Case 1 and 2 had positive titer for precipitation IgG antibody to cow’s milk but was not measured in case 3, so I have added this to the text and Table 1. (page 1, line 35-37).
“Diagnosis is made by a combination of clinical, laboratory including positive precipitating IgG antibody to cow’s milk, and radiographic evidence. But most importantly, clinical exacerbation of pulmonary symptoms or chest radiograph after cow’s milk consumption is pathognomonic”
However, while laboratory test may provide a helpful clue in diagnosing Heiner syndrome, it is not pathognomic, and we wanted to emphasize the importance of clinical diagnosis of Heiner syndrome. As also mentioned in a recent journal by Arasi S et al. in 2021 entitled “Heiner Syndrome and Milk Hypersensitivity: An Updated Overview on the Current Evidence,” the proposed diagnostic approach is made probable when (A) pulmonary symptoms and chest x-ray infiltrates or pulmonary hemosiderosis; and (B) resolution after milk restriction; and convincing if also (C) recurrence after milk reintroduction, which our case report also supports. Milk precipitin as well as serum IgE and/or allergic sensitization to milk as positive skin prick test are all tests that may be positive that can hint and favor the diagnosis of Heiner syndrome but if clinically does not exacerbate at the exposure of milk, patient is less likely to be Heiner syndrome. So the tests are some supplementary information to be considered in our point of view, so was not done in Case 3.
Reviewer 3 Report
The case report „ Children with Heiner syndrome: A single-center experience” highlights the role of Cow Milk Allergy in rare conditions in children. The authors also highlighted the role of eviction and reintroduction of Cow Milk in those cases. I have read the paper with interest and feel that it is relevant for area of Food Allergies.
I suggest few minor revisions and comments are made below regarding the article.
- The authors should clarify for all the patients if there were any additional symptoms present like: failure to thrive, colic behavior, vomiting, hematochezia, diarrhea, recurrent fever, etc.
- In the Laboratory test the authors should clarify for all the patients: if IgG to cow milk were determined, if blood eosinophils were increased and what type of substitution for cow milk was introduced. Also, add in discussion those components.
- In the case 2. the authors should clarify if infections were excluded and explain the frequency and the reason for accidental exposure to cow milk if it is possible.
- In Discussion: Add more discution about pathophysiology and management. The authors may read:
Arasi S, Mastrorilli C, Pecoraro L, Giovannini M, Mori F, Barni S, Caminiti L, Castagnoli R, Liotti L, Saretta F, Marseglia GL, Novembre E. Heiner Syndrome and Milk Hypersensitivity: An Updated Overview on the Current Evidence. Nutrients. 2021 May 18;13(5):1710. doi: 10.3390/nu13051710. PMID: 34070007; PMCID: PMC8157832.
Author Response
The case report, Children with Heiner syndrome: A single-center experience” highlights the role of Cow Milk Allergy in rare conditions in children. The authors also highlighted the role of eviction and reintroduction of Cow Milk in those cases. I have read the paper with interest and feel that it is relevant for area of Food Allergies.
I suggest few minor revisions and comments are made below regarding the article.
Point #1: The authors should clarify for all the patients if there were any additional symptoms present like: failure to thrive, colic behavior, vomiting, hematochezia, diarrhea, recurrent fever, etc.
Response: Our first case exhibited mild fever for three days, which allowed him to go to local clinic to take a blood test and be diagnosed with anemia, but had no prior history of recurrent fever or respiratory infection. Case 3 had additionally failure to thrive with short stature (5-10 percentile) and low weight below 3rd percentile compared to his age upon diagnosis, so and after cessation of corticosteroid treatment, he is currently 5-10 percentile. Case 2 did not exhibit additional symptoms.
Point #2: In the laboratory test the authors should clarify for all the patients: if IgG to cow milk were determined, if blood eosinophils were increased and what type of substitution for cow milk was introduced. Also, add in discussion those components.
Response: Milk precipitin IgG levels were added to Table 1, and were positive in case 1 and 2, while it was not measured in case 3. Blood eosinophils were not elevated in all three cases. Cow’s milk substitution was not done since they were all above age that required milk as the main diet but soybean formula could have been considered as substitute.
Point #3: In the case 2. the authors should clarify if infections were excluded and explain the frequency and the reason for accidental exposure to cow milk if it is possible.
Response: As you can see in Table 1, infection markers were excluded in common for all three cases since it is the first differential diagnosis to be made when chest x-ray is seen with diffuse pulmonary infiltrate. The “accidental” exposure to cow’s milk was due to carelessness of the parents who kept it around the child, so she couldn’t help drinking it. Although accidental, we regarded it as inadvertent oral provocation test since the patient developed symptoms after exposure.
- In Discussion: Add more discussion about pathophysiology and management. The authors may read:
Arasi S, Mastrorilli C, Pecoraro L, Giovannini M, Mori F, Barni S, Caminiti L, Castagnoli R, Liotti L, Saretta F, Marseglia GL, Novembre E. Heiner Syndrome and Milk Hypersensitivity: An Updated Overview on the Current Evidence. Nutrients. 2021 May 18;13(5):1710. doi: 10.3390/nu13051710. PMID: 34070007; PMCID: PMC8157832.
Response: Thank you very much for your suggested reading. I have added the following text about the pathophysiology of Heiner syndrome in the discussion section.(page 4, line 123-126)
“Moreover, there are many hypotheses regarding cow’s milk hypersensitivity in Heiner syndrome such as immune complex formation against cow’s milk protein, deposition of complement, and involvement of immunoglobulin, fibrin, and milk protein, and so on, but the exact mechanism remains unclear.
Round 2
Reviewer 1 Report
I can confirm that the parts I pointed out have been improved.